# Phenolic Profile and Antioxidant, Anticholinergic, and Antibacterial Properties of Corn Tassel

**DOI:** 10.3390/plants11151899

**Published:** 2022-07-22

**Authors:** Jameel M. Al-Khayri, Arzu Kavaz Yüksel, Mehmet Yüksel, Mesut Işık, Emrah Dikici

**Affiliations:** 1Department of Agricultural Biotechnology, College of Agriculture and Food Sciences, King Faisal University, Al-Ahsa 31982, Saudi Arabia; 2Department of Food Technology, Technical Sciences Vocational School, Atatürk University, Erzurum 25240, Turkey; 3Department of Food Engineering, Faculty of Agriculture, Atatürk University, Erzurum 25240, Turkey; mehmet.yuksel@atauni.edu.tr; 4Department of Bioengineering, Faculty of Engineering, Bilecik Şeyh Edebali University, Bilecik 11230, Turkey; mesut.isik@bilecik.edu.tr; 5Science and Technology Application and Research Center, Aksaray University, Aksaray 68100, Turkey; emrah.dikici25@gmail.com

**Keywords:** corn tassel (CT), antioxidant, phenolic compounds, acetylcholinesterase, antibacterial

## Abstract

Corn tassel (CT) is a waste part of the corn plant. It is a good co-product and rich in terms of bioactive compounds and phytochemicals. This research tried to show the phenolic profile, antioxidants, anticholinergic activities, and antibacterial properties of CT ethanol extract. The phenolic content analysis of the CT was determined quantitatively by LC-MS/MS, and the antioxidant capacity was measured using ABTS, DPPH, Cu^2+^–Cu^+^, and Fe^3+^–Fe^2+^ reducing methods. The anticholinergic measurements of CT were detected by inhibition of acetylcholinesterase (AChE). The antibacterial activity was determined by MIC and disc diffusion methods. Many phenolic compounds such as vanillic acid, caffeic acid, fumaric acid, acetohydroxamic acid, butein, myricetin, resveratrol, catechin hydrate, and 4-hydroxybenzoic acid were detected in ethanol extract of CT. The obtained plant ethanol extract had a 7.04% DPPH value, while it showed ABTS activity at 9.45%. Moreover, it had a 0.10 mg/mL inhibition effect on the AChE in terms of IC_50_ values. The ethanol extract of the CT had an antibacterial property on the investigated bacteria at different ratios. In conclusion, this research aims to consider CT as a source of phenolic compounds and to reveal its bioactive properties and its effects on the treatment of some diseases.

## 1. Introduction

Corn (*Zea mays* L.) is one of the most produced cereals in the world. The various uses of corn as food, fodder, animal feed, and lately biofuel have further made it a more favorable and valuable crop. Worldwide, on 197.20 million hectares of land, 1148.48 million metric tonnes of maize were harvested, on average [1].

Corn kernels are an important material of the corn plant and are used as a raw material in starch and flour making, production of animal feed, and industrial products such as bioethanol, enzymes, vitamins, organic acids, etc. Other important waste products derived from the corn plant are corn tassel, kernel, and corncobs, which are good sources of bioactive compounds such as flavonoids, phenolic compounds, anthocyanins, proteins, vitamins, minerals, carotenoids, and volatile compounds [2,3,4,5].

Corn tassel (CT) takes place at the top of the corn plant as a male floral organ and produces pollens being carried by the wind to pollinate female flowers named silk. Tassel may produce an average of 25 million pollen grains [6]. Pollens have a high potential for health and are a good source of nutrients and antioxidants, proteins, oils, and sugars [7].

An average of 25–30 g of the fresh tassel is obtained from a corn plant, and its weight after drying is 8–9 g at most. The weight of tassels in corn depended on the structure of tassels in different varieties, especially the number of tassel branches, the length of anthers and tassels, and other factors [5].

CT is a waste part of the corn plant, and calcium is considered an important renewable raw material for industry. It has many bioactive compounds, especially calcium, potassium, sodium, magnesium, proteins, vitamins, carbohydrates, lipids, minerals, volatile oil compounds, steroid compounds (sitosterol and stigmasterol), saponin, tannin, alkaloid, and flavonoids that cause useful effects on people’s health [7,8,9]. Furthermore, it is a valuable co-product in terms of phytochemicals, glycosides of quercetin, isorhamnetin, and kaempferol [10,11,12,13,14,15]. Traditionally, CT has been used as uricosuric, antiseptic, diuretic, and antilithiasic. In addition, it is evaluated among people for the healing of diseases such as cystitis, gout, prostatitis, edema, kidney stones, and nephritis [16].

Phenolic compounds found in CT have an important effect, such as anti-atherogenic, antimicrobial, antiallergenic, antioxidant, cardioprotective, anti-inflammatory, and antithrombotic effects [17]. In addition, it can prevent the risk of some cancers, cardiovascular disease, diabetes obesity, and digestive problems [18,19,20,21,22,23,24]. On the other hand, CT extract causes an increase in insulin levels and healed injured β cells [25,26].

CT shows more effectively, especially in kidney diseases such as benign prostate hyperplasia, chronic nephritis, cystitis, and gout. It helps to pass through kidney stones from the kidney and urinary tract. Moreover, CT shows anti-prostatitis and antispasmodic properties [14,27].

CT is used medicinally against various pathogenic bacteria that cause certain diseases. It has antimicrobial properties that prevent some diseases stemming from bacteria, fungus, and viruses. The plant-originated antimicrobial factors have the least side effects on health and show prospective therapeutic effects on many infectious diseases [28].

In previous studies, the phenolic acid content of natural-origin CT has not been studied. On the other hand, previous studies mostly focused on the antibacterial effect of corn silk extracts, while antibacterial activity against the CT extracts was not evaluated [29,30]. Furthermore, there is no study about the determination of some bioactive compound profiles and antibacterial properties of CT grown in the Trabzon region. This study aims to show phenolic compound composition and some other important properties of CT grown in Trabzon, Turkey. In addition, CT’s antioxidant, anticholinergic, and antibacterial properties were determined using different types of methods. In addition, this study aims to bring together some beneficial information about CT.

## 2. Results and Discussion

### 2.1. The Phytochemical Phenolic Compound Profiles of CT

Phenolic compounds of natural origin that occur in plants have many biological functions. They can inhibit the various stages of cancers and show protection against cardiovascular disease [31]. Moreover, phenols have protective effects against lipoprotein oxidation [32]. Flavonoids are phenolic compounds that are present in vegetables, fruits, flowers, and barks [33]. It has important biological activities including antiviral, antibacterial, anti-allergic, and anti-tumor [34,35].

In this study, the contents of more than twenty phenolic compounds commonly found in plants were investigated. The chromatogram data for phenolic acids in CT, linear regression equations, and linearity ranges of standard phenolic compounds were given in Table 1 and Figure 1. The correlation coefficients (R^2^) of the standards were found around 0.994–1.011. For this analysis, the LOQ value was between 0.1 and 214.3 μg/L, while the LOD value was in the range of 0.5–206.8 μg/L. When the method validation studies were examined in the literature, it might be said that the determined values showed parallelism with the findings in this study [36]. In a study, it was determined that the total phenolic compounds amount extracted from CT using different solvents was found to be lower than in ethanol extracts. Therefore, ethanol for CT was preferred as the solvent in the phenolic acid content analysis [37].

The phenolic compound profiles of CT were determined compared to standard compounds. As seen in Table 1, the major compound of CT was detected as vanillic acid and followed by caffeic, fumaric, and acetohydroxamic acids, butein, myricetin, resveratrol, catechin hydrate, and 4-hydroxybenzoic acid, respectively. The determined compounds consisted of flavonoid glycosides, phenolics, phenolic acids, and their derivatives. Khider et al. [38] reported that vanillic acid, *p*-coumaric acid, protocatechuic acid, anthocyanins, derivatives of hesperidin and quercetin, and *p*-coumaric and ferulic acid were found in corn silk. Kapcum et al. [9] detected vanillic, caffeic, ferulic, *p*-coumaric, *p*-hydroxybenzoic, and sinapic acids, as well as kaempferol and quercetin, in different parts of purple waxy corn. Additionally, del Pozo-Insfran et al. [39] stated that the polyphenolic compound profiles and amounts might be different depending on their properties, origin, and growing environment.

### 2.2. Antioxidant Properties of CT

The phenolic compounds have antioxidant properties due to their structure. These effects of theirs stem from their chelating the metal ions, free radical scavenging ability, or donating hydrogen atom electrons. All parts of the corn are an important source of different phytochemical compounds. CT includes phenolic acids, flavonoids, glycosides, anthocyanins, polyphenols, polysaccharides, and carotenoids [40].

The free radical scavenging activity of CT ethanol extracts was found with DPPH and ABTS methods. Nurhanan et al. [41] reported that the DPPH radical scavenging activity of corn silk methanol and water extracts was determined as 80% and 60% (1 mg/mL), respectively.

As seen in Table 2, 0.2 mg/mL CT extract showed 7.04 ± 0.20% DPPH and 9.45 ± 1.20% ABTS radical scavenging activities. Trolox had the highest (83.19 ± 4.00%) antioxidant activity, which was followed by BHA (72.84 ± 4.80%), BHT (47.32 ± 2.61%), and CT extract (7.04 ± 0.20%), respectively (Table 2). In one study, it was stated that caffeic acid had high DPPH radical scavenging potential with IC_50_ values of 6.46 ± 0.17 μM [42]. Furthermore, the vanillic acid and caffeic acid in CT were found to be higher than in other phenolic acids. From these results, it can be said that the radical scavenging activities might be due to these phenolic acids.

The antioxidant activities of the CT extract were found to be lower than the standards. These results might be caused by the presence of a low amount of antioxidant compounds in CT. The determined DPPH scavenging values showed that the CT extract has free radicals scavenging activity. CT also might show preventing properties on some free radical-mediated chain reactions as a potential source of natural antioxidants. Ku et al. [43] reported that antioxidant activities of the corn silk showed differences depending on the variety and type of corn and its maturity at the time of harvest. In addition, Nurhanan et al. [41] reported similar results to our study in corn silk extracted with methanol (81.7% at 1000 μg/mL).

The CUPRAC method is an effective, simple, rapid, and selective method for a wide spectrum of polyphenolic substances [44]. The high potential of CT for free radical scavenging activities in DPPH assays might be correlated with the phytochemicals and total phenolic content [45]. The important property of antioxidant reactions is the deactivation or chelating of the metals that are likely to catalyze hydroperoxide decomposition and Fenton-type reactions [46]. The ABTS method stems from the capabilities of antioxidants to donate an electron or hydrogen atom to stabilize radicals and thus convert them to non-radical species [47]. Table 2 showed that the highest metal-reducing activity in CUPRAC and Fe^3+^–Fe^2+^ reducing activity values were determined for BTH, while the lowest activity was found in CT extract. From these findings, it might be said that the CT extract showed lower iron and copper reducing activity than BHA, BHT, and Trolox. The phenolic components found in CT had an important metal-reducting and radical removal capacity.

Vanillic, coumaric, and especially ferulic acid can inhibit the radical-induced toxicity, reducing OH·-induced oxidation of membrane lipids and proteins. In this study, it can be predicted that the antioxidant content of CT might be caused by phenolic compounds, especially vanillic acid. Although the results showed that the ethanolic CT extract had low antioxidant power, it might be successfully applied in food processing instead of synthetic compounds due to its naturally produced phenolic compound content [46,47].

### 2.3. Anticholinergic Effect of CT

Compounds with anticholinergic effects are utilized extensively for the treatment of peptic ulcers and Alzheimer’s disease (AD) [48]. According to the cholinergic hypothesis, the decrease in the neurotransmitter acetylcholine (ACh) level causes the formation of AD. The inhibitors of AChE are used to treat symptomatic AD that raises the antioxidant substance development and thus protects the cells against oxidative destructions. This neurological illness causes important degeneration in tissues of the brain, due to acetylcholine (ACh) deficiency, and affects behavior and memory, especially in elderly individuals [49]. The decreasing levels of ACh in the cortex and hippocampus cause great changes in AD patients [50,51,52]. Natural products that are rich in terms of phenolics and AChE inhibitors have been generally used for the treatment of AD [53]. There is a lot of important development in the treatment of AD disease using the AChE inhibitors.

Table 2 showed that ethanolic extract of CT had IC_50_: 0.10 ± 0.01 mg/mL value. Anwar et al. [54] reported that caffeic acid changed AChE activity in different tissues in vitro and in vivo. The caffeic acid inhibited AChE activity in muscles in vitro at different concentrations between 0.5 and 2 mM, while activation of the AChE was observed in rats when treated with caffeic acid in the range of 10 to 100 mg/kg. In another study, AChE inhibitor potentials were investigated by extracting many plant species with different solvents. It was found that plant extracts have an AChE inhibitory potential of approximately 30% to 95% at 1 mg/mL in different solvents, while the ethanol extract of CT had a 50% effect at 0.1 mg/mL [55]. Medicinal aromatic plant-derived phenolic compounds have anticholinergic effects and can be used as an alternative AChE potential inhibitor to synthetic drugs in the treatment of AD.

### 2.4. Antibacterial Properties of CT

The antibacterial properties of CT extract were determined against *S. typhimurium*, *S. aureus*, and *E. coli* as shown in Table 3 and Table 4.

Table 3 showed that the 312 μg/mL CT extract had the highest and same inhibition zone diameter on *E. coli* (6.0 ± 0.10 mm) and *S. typhimurium* (6.0 ± 0.22 mm), while its influence was measured as 4.0 ± 0.10 mm on *S. aureus*. The antibacterial effect of CT extract on studied bacteria cultures was determined to be quite low compared to the ciprofloxacin antibiotic. Eman [56] reported that corn silk has no antibacterial activity against investigated bacterial species (*Staphylococcus aureus*, *Streptococcus pneumonia*, *Escherichia coli*, *Pseudomonas aeruginosa*, *Klebsiella pneumonia*, and *Streptococcus pyogenes*). Nessa et al. [29] found corn silk or tassel extracts had an antibacterial effect against *E. coli and S. aureus*. Similarly, Salar et al. [57] reported that corn silk and tassel extracts had antimicrobial effects on *Salmonella* species.

The 10 μL bacterial inoculum was applied to 312 μg/mL, 156 μg/mL, 78 μg/mL, 39 μg/mL, 19.5 μg/mL, and 9.75 μg/mL concentrations of CT extract for the determination of Minimum Inhibitory Concentration (MIC) results (Table 4). Only the 312 μg/mL plant extract concentration showed an inhibition effect on *E. coli* and *S. typhimurium*, while this concentration did not affect *S. aureus*. Other concentrations did not produce any inhibition effect on the studied bacteria. These results revealed that the antibacterial activity of CT is very low. Many phenolic compounds have also significant antimicrobial activities apart from their antioxidant activity. Mohsen et al. [58] reported that the natural extracts of corn have antioxidant agents as well as their property of inhibiting fungi-producing toxins.

## 3. Materials and Methods

### 3.1. Preparation of Plant Extract Materials 

In this study, CT on the top of the plant was used as the material (Figure 2). The CT was collected in August 2021 from Trabzon, Türkiye (Latitude: 39.58. Longitude: 40.80, Elevation: 398 m/1306 feet). We collected fresh tassels, on average 25 g for each corn, and obtained the highest corn tassel dry weight as 8–9 g. The obtained CT (2.50 kg fresh CT) was dried for 10–15 days at room temperature. Then, obtained materials were stored in jars at −4 °C for analyses. The chemicals used were purchased commercially, and analyses were performed in triplicate.

### 3.2. Plant Extract Preparation

Atmani’s [59] method was used for the extraction of CT. The mixture of sample: ethanol (1:10) was shaken for 24 h. The obtained mixture was filtered with filter paper and evaporated. The dried material was stored at 4 °C.

### 3.3. Phenolic Compound Analysis

The phenolic acid content analysis in CT was performed according to the method developed by Yilmaz [60]. Calculations were made with Lab Solutions software (Shimadzu, Kyoto, Japan). The phenolic compound analyses of CT ethanol extract were made by LC-MS/MS (Nexera model Shimadzu UHPLC and a tandem MS device). The CTO-10ASVP column furnace, DGU-20A3R degasser LC-30AD dual pumps, and SIL-30AC autosampler were used for the phenolic analyses. Chromatographic separation of compounds was made using a C18 Inertsil ODS-4 (3.0 mm × 100 mm, 2 µM) column. The column temperature was determined as 40 °C. The elution gradient was created using mobile phase A (0.1% formic acid and water) and mobile phase B (0.1% formic acid and methanol). The injection value was adjusted to 4 µL, and the solvent flow ratio was 0.5 mL/min [61]. The working method of LC/MSMS is shown in Table 5.

### 3.4. Antioxidant Properties

#### 3.4.1. Total Reduction Capability

The metal reduction properties of CT materials have been performed by a method found by Elmastaş et al. [62]. The different concentrations of ethanolic extracts (10, 20, 40 µL of CT ethanol extract taken from 1 mg/mL stock solution) were mixed with 2.5 mL 1% potassium ferricyanide and 2.5 mL phosphate buffer (0.2 M, pH 6.6). Then they were incubated at 50 °C for 15–20 min. The 2.5 mL 10% TCA and 0.25 mL, 0.1% FeCl_3_ were transferred to all mixtures and centrifuged for 10 min at 3000 rpm. These absorbances were determined at 700 nm for mixtures. The determined results were compared with standards.

#### 3.4.2. Cu^2+^ Reduction Capacity (CUPRAC)

The basis of this method is the reduction of Cu (II)-Nc to Cu (I)-Nc [63]. A total of 1 mL NH_4_Ac buffer solution, 1 mL CuCl_2_ (0.01 M) solution, and 1 mL neocuproine (2,9-dimethyl-1,10-phenanthroline) was mixed with different concentrations of CT ethanol extracts. The obtained total quantity was completed to 4 mL with ultrapure water. The absorbances of the samples were measured at 450 nm after holding them at room temperature for 30 min. The found results were given as absorbance and compared to standard antioxidants.

#### 3.4.3. DPPH Removal Activity

The DPPH values of CT ethanol extracts and standards were determined using the Blois method [64]. The 0.1 mM DPPH solution was prepared using methanol. The 10, 20, and 40 µL of samples (taken from 1 mg/mL stock solution) were mixed with 1 mL of DPPH solution and completed to 3 mL with ethanol, and then vortexed. The mixture was incubated for 30 min in the dark. The absorbance of the samples was read at 517 nm.

#### 3.4.4. ABTS Removal Activity

The ABTS analysis relies on the color change after the mixture of colored ABTS^+^ cation radical with plant extracts [65]. The 2 mmol/L ABTS solutions were mixed with 2.45 mmol/L potassium persulfate and kept on hold at room temperature for 14 h in dark conditions. The ABTS+ radical solution was mixed with sodium phosphate buffer (0.1 mol/L, pH 7.4) until reaching an absorbance of 734 nm. Then, 10, 20, and 40 µL of CT ethanol extract (1 mg/mL, stock solution) were completed to 3 mL with phosphate buffer. The prepared 1 mL of ABTS^+^ solution was mixed with the samples and then vortexed. The absorbance of the mixtures was read at the absorbance of 734 nm using a spectrophotometer.

### 3.5. Acetylcholinesterase (AChE) Activity

The Ellman [66] method was used for the determination of the CT ethanol extract on the AChE enzyme. The reaction solution (50 µL, (DTNB), 100 µL Tris–HCl buffer (1 M, pH 8.0), and 50 µL AChE (5.32 × 10^−3^ U)) was kept at 30 °C and then stirred (approximately 15 min). The reaction was started with the addition of 50 µL of AChE as a substrate, and enzymatic hydrolysis of it was detected at 412 nm. The IC_50_ values were given from activity (%)-(Ligand) graphs for extract [67].

### 3.6. Determination of the Antibacterial Activity of CT Extract

*S. typhimurium* (ATCC 14028), *E. coli* (ATCC 25922), and *S. aureus* (ATCC 25923) have used the determination of the antibacterial effects of CT extracts. The cultures were activated in Trypticase Soy Yeast Extract at 35–37 °C for 24 h. The densities of bacteria were set to 1 × 10^8^ CFU/mL by comparing 0.5 McFarland [68].

#### 3.6.1. Disc Diffusion Method

The 20 mL of Mueller–Hinton agar was added to a 9 cm diameter of sterile Petri dishes. Each bacterial suspension (standard quantities (10^8^ CFU/mL)) was transferred to a Petri. Then, 20 µL of CT extract impregnated with 6 mm of sterile paper discs were placed on a medium. The prepared Petri dishes were kept at 37 °C for 24–48 h. The diameter of the bactericidal zones (mm) was measured [69].

#### 3.6.2. Minimum Inhibitory Concentration (MIC) Method

The CT extracts were dissolved in dimethyl sulfoxide (DMSO; 35%) at 312 mg/mL (*w*/*v*) concentration. The solutions were sterilized with 0.45 μm filters (Millipore, France). The sterile extracts were stored in 1.5 mL microtubes in the refrigerator (4 °C). Bacterial inoculums (10 μL) were added to the microwell and the CT extract was diluted to 312, 156, 78, 39, 19.5, 9.75 mg/mL with nutrient broth (NB). The sterile DMSO solution was used as a negative control. The microwell plates were incubated at 35 °C–37 °C for 24 h. The turbidity in the wells and invisible growth in the Petri dishes were accepted as positive activities. The analyses were performed in triplicate [69].

### 3.7. Statistical Analysis

The analysis results were evaluated with GraphPad Prism version 8 (GraphPad Software, La Jolla, CA, USA). The results were shown as 95% confidence intervals (mean ± standard deviation).

## 4. Conclusions

The obtained results demonstrated that the CT extracts had a metal reduction, free radical scavenging anticholinergic, and antibacterial activities. This plant part is promising for the treatment of important health problems due to its various features. The antioxidants and phenolics that are extracted from CT can be used as a functional food additive, in cosmetic products, natural pharmaceuticals, and nutraceutical industries. In addition, CT and obtained products can be used as a future food supply for humans.

## Figures and Tables

**Figure 1 plants-11-01899-f001:**
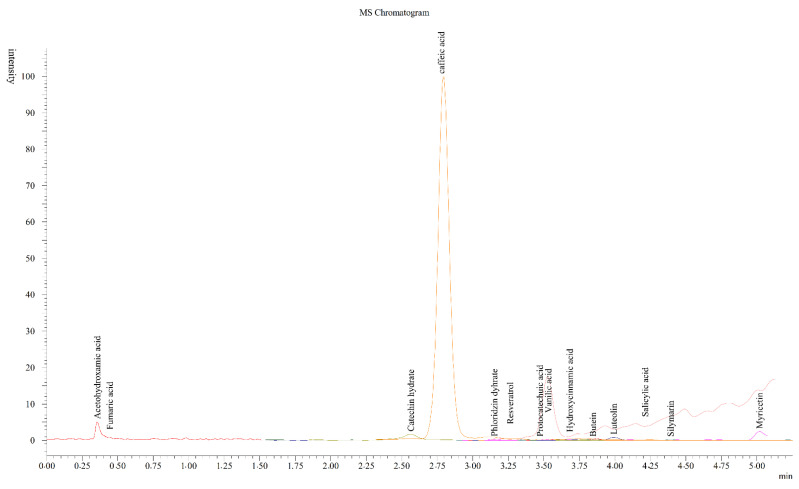
The chromatogram of the phenolic acids in CT extract obtained by the LC-MS/MS method.

**Figure 2 plants-11-01899-f002:**
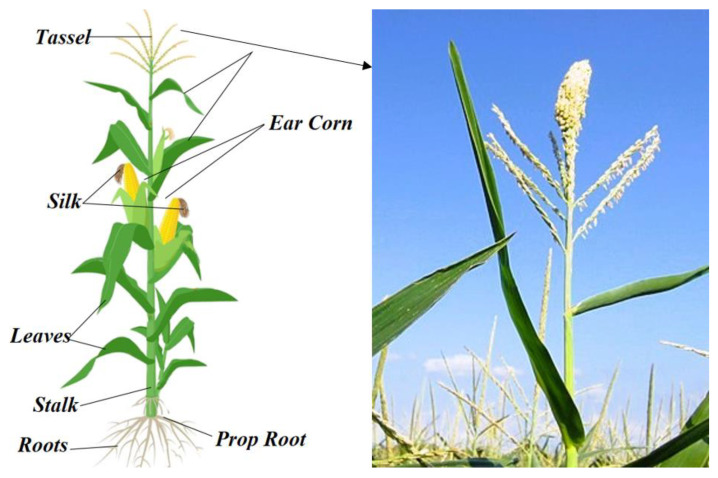
Parts of *Zea mays* L. and corn tassel.

**Table 1 plants-11-01899-t001:** The determination of phytochemical profiles of CT.

Standard Compounds	Max Absorbance (λmax, nm)	^a^ MRM	^b^ RSD %	^c^ LOD/LOQ (μg/L)	Recovery (%)	^d^ RT	^e^ R^2^	Equation	Concentration (µg/L)
Quercetin	254	301.1 > 151	0.0136	22.5/25.7	1.001	3.891	0.999	Y = (13.7831)X + (−146.951)	Not detected
Acetohydroxamic acid	502	76.10 > 43.10	0.0082	2.8/8.2	1.000	0.355	0.999	Y = (150.982)X + (23.1833)	145.64
Catechin hydrate	278	291.10 > 139.00	0.0236	8.2/11.4	0.994	2.564	0.999	Y = (79.2933)X + (−2406.22)	15.39
Vanillic acid	260	168.80 > 93.00	0.0062	125.5/142.2	1.001	3.518	0.998	Y = (48.0522)X + (−876.904)	1046.26
Resveratrol	288	229.10 > 135.00	0.0131	9.0/13.6	0.998	3.188	0.998	Y = (46.4361)X + (−1314.61)	20.27
Fumaric acid	365	115.20 > 71.00	0.0047	25.2/31.3	0.997	0.500	0.999	Y = (20.2986)X + (−762.592)	151.51
Gallic acid	288	169.20 > 125.00	0.0136	0.90/1.6	1.000	1.442	0.999	Y = (65.3835)X + (−2699.84)	Not detected
Caffeic acid	330	179.20 > 135.00	0.0137	6.3/10.7	1.009	2.770	0.996	Y = (124.785)X + (−487.132)	669.71
Phloridzin dihydrate	254	435.00 > 273.10	0.0564	61.0/207.0	1.000	3.174	0.999	Y = (33.4069)X + (−1396.90)	Not detected
Oleuropein	278	539.10 > 377.20	0.0694	0.05/1.0	0.997	3.567	0.999	Y = (25.9240)X + (−558.916)	Not detected
Ellagic acid	259	300.90 > 145.10	0.0856	0.101/0.333	1.002	3.861	0.999	Y = (5.25903)X + (−1167.31)	Not detected
Myricetin	330	317.10 > 150.90	0.0079	55.4/59.6	0.999	5.017	0.999	Y = (37.0934)X + (2684.23)	29.36
Protocatechuic acid	280	181.20 > 108.00	0.0129	30.3/35.4	1.011	3.556	0.994	Y = (526.954)X + (23,026.1)	Not detected
Butein	378	271.10 > 135.00	0.0145	22.7/28.6	0.096	3.847	0.999	Y = (49.3543)X + (367.917)	59.65
Naringenin	288	271.10 > 150.90	0.0205	5.4/6.4	0.998	3.952	0.996	Y = (317.241)X + (33,733.3)	Not detected
Luteolin	330	285.20 > 132.90	0.0057	0.5/2.5	1.007	3.982	0.999	Y = (34.6668)X + (3721.79)	Not detected
Kaempferol	265	285.10 > 116.90	0.0144	206.6/214.3	0.999	3.928	0.999	Y = (2.63905)X + (−206.494)	Not detected
Alizarin	430	239.20 > 210.90	0.0351	65.2/77.5	0.966	4.594	0.999	Y = (3.97487)X + (1614.23)	Not detected
4-Hydroxybenzoic acid	278	137.20 > 93.00	0.0154	30.5/40.25	0.996	3.528	0.999	Y = (735.804)X + (−498.102)	10.52
Salicylic acid	278	137.20 > 93.00	0.0124	4.2/7.6	1.009	4.212	0.999	Y = (746.369)X + (6072.41)	Not detected

^a^ MRM: Multiple reaction monitoring. ^b^ RSD: Relative standard deviation. ^c^ LOD/LOQ (µg/L): Limit of detection/ limit of quantitation. ^d^ RT: Retention time. ^e^ R^2^: Determination coefficient.

**Table 2 plants-11-01899-t002:** The antioxidant and AChE inhibition properties of CT.

Samples	DPPH ^a^ (0.2 mg/mL)	ABTS ^a^ (0.2 mg/mL)	FRAP ^b^ (0.2 mg/mL)	CUPRAC ^b^ (0.2 mg/mL)	AChE	
					IC_50_ (mg/mL)	R^2^
CT	7.04 ± 0.20	9.45 ± 1.20	0.13 ± 0.01	0.19 ± 0.01	0.10 ± 0.01	0.947 ± 0.02
BHA	72.84 ± 4.80	84.60 ± 3.00	0.44 ± 0.02	0.60 ± 0.01		
BHT	47.32 ± 2.61	50.32 ± 3.20	0.62 ± 0.02	0.64 ± 0.01		
Trolox	83.19 ± 4.00	81.00 ± 5.31	0.27 ± 0.01	0.53 ± 0.01		

Standards (BHA, butylated hydroxyanisole; BHT, butylated hydroxytoluene; and Trolox). ^a^ Results are given as the percent radical scavenging properties. ^b^ Results are given as absorbance.

**Table 3 plants-11-01899-t003:** The antibacterial properties of CT (diameter inhibition; mm).

Sample (6.24 μg/disk)	Concentration	*S. aureus*	*E. coli*	*S. typhimurium*
CT	312 μg/mL	4.0 ± 0.10	6.0 ± 0.10	6.0 ± 0.22
Control (Ciprofloxacin)	5 μg/mL	19.0 ± 0.12	21.0 ± 0.12	16.0 ± 0.20

**Table 4 plants-11-01899-t004:** MIC (minimum inhibitory concentration) results of CT extract.

CT	Concentration (μg/mL)	Inoculum Amount (μL)	*S. aureus*	*E. coli*	*S. typhimurium*
	312	10	+	−	−
	156	10	+	+	+
	78	10	+	+	+
	39	10	+	+	+
	19.5	10	+	+	+
	9.75	10	+	+	+
Medium + Inoculum	0	10	+	+	+
Medium + DMSO	0	10	+	+	+
Medium	0	0	−	−	−

(+): Development, (−): No development.

**Table 5 plants-11-01899-t005:** The working method of LC/MSMS.

**Mobile Phase**	A: 0.1% formic acid and water, B: 0.1% formic acid and methanol
**Flow Rate**	0.5 mL/min
**Column Temperature**	40 °C
**Column Property**	C18 Inertsil ODS-4 (3.0 mm × 100 mm, 2 µM)
**Injection Volume**	4 µL

## Data Availability

Not applicable.

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
