# Peer review of "Phenolic Profile and Antioxidant, Anticholinergic, and Antibacterial Properties of Corn Tassel"

_plants, 2022, doi:10.3390/plants11151899_

Round 1
Reviewer 1 Report
The works of Al-Khayri, focus on the Phenolic Profile, Antioxidant, Anticholinergic, and 2 Antibacterial Properties of Corn Tassel, needs to be significantly improved in order to be considered for publication.
Main concerns:
- In introduction, the authors must clearly state the state of the art (actual knowledge of previous reports) and emphasize the novelty of their work.
- Improve material and methods section. Eg the method/operation conditions of LC-MS is not clear; Validation of the method is also missing; as CT was obtained in the dried form, please indicate the solvent to solubilize it for analysis of 4.3 – 4.6. Clarify concentrations (final concentrations) used in the analysis (eg section 4.4.3/DPPH method it is stated that 40, 20, and 10 μg/mL of samples were used … but only a concentration ? But in section 2.2 only a concentration (0.2 ug/mL) is mentioned)?
- UV spectral data of the identified compounds must be added in the table 1.
- Chromatogram of the sample should be shown
- The antioxidant activity of the sample, as measured by the applied methods, is negligible. This must be clear in all parts of the manuscript
- In general, the paper does not contain a discussion (comparison with other works) related with the same kind of samples/plant or the same botanical species….
Author Response
Manuscript number: plants-1799125
MS Type: Full Paper
Title: Phenolic Profile Antioxidant, Anticholinergic, and Antibacterial Properties of Corn Tassel
Correspondence Authors: Prof. Dr. Jameel M. Al-Khayri and Assoc. Prof. Arzu Kavaz Yüksel
Response to Reviewer Comments
First, I want to thank you for your informative and necessary comments on my manuscript. I have tried to do necessary modifications in light of your recommendations accordingly and detailed corrections are listed below point by point.
All manuscript was revised based on the editor and reviewer’s comments. The corrections were done in the text according to the reviewer’s suggestion. The changes made during the manuscript revision have been highlighted with a yellow background for the REVIEWER1 comment; while REVIEWEr 2's comments were highlighted with a green background and general changes were highlighted in gray.
We appreciate your time and recommendations to improve this paper.
Thank you for your interest.
Reviewer #1:
Q1. In the introduction, the authors must clearly state the state of the art (actual knowledge of previous reports) and emphasize the novelty of their work.
R1. The necessary corrections were done in the text according to the reviewer’s suggestion and the introduction section has been reviewed and revised.
Q2. Improve the material and methods section. Eg the method/operation conditions of LC-MS are not clear; Validation of the method is also missing.
R2. In this study, no validation study was performed for LC/MSMS analyzes, but it was carried out on previously developed validated parameters (Yilmaz, M. A. (2020) reported simultaneous quantitative screening of 53 phytochemicals in 33 species of medicinal and aromatic plants: A detailed, robust and comprehensive LC-MS/MS method validation. Industrial Crops and Products, 149, 112347.). The working method of LC/MSMS showed in Table 5. Necessary changes were made to the manuscript.
Q3. CT was obtained in the dried form, please indicate the solvent to solubilize it for analysis of 4.3 – 4.6. Clarify concentrations (final concentrations) used in the analysis (eg section 4.4.3/DPPH method it is stated that 40, 20, and 10 μg/mL of samples were used … but only a concentration? But in section 2.2 only a concentration (0.2 ug/mL) is mentioned)?
R3. In the study, the sample concentrations are given as 10, 20, and 40 μg/mL inadvertently, and these values are the volumetric values taken from the 1 mg/mL stock sample. (Necessary corrections have been made in the text and highlighted in yellow).
Q4: UV spectral data of the identified compounds must be added in Table 1.
R4. UV spectral data are presented in Table 1.
|
Standard compounds |
Max Absorbance (λmax) |
|
Quercetin |
254 nm |
|
Acetohydroxamic Acid |
502 nm |
|
Catechin hydrate |
278 nm |
|
Vanillic Acid |
260 nm |
|
Resveratrol |
288 nm |
|
Fumaric Acid |
365 nm |
|
Gallic acid |
278 nm |
|
Caffeic Acid |
330 nm |
|
Phloridzin dihydrate |
254 nm |
|
Oleuropein |
278 nm |
|
Ellagic Acid |
259 nm |
|
Myricetin |
330 nm |
|
Protocatechuic acid |
280 nm |
|
Butein |
378 nm |
|
Naringenin |
288 nm |
|
Luteolin |
330 nm |
|
Kaempferol |
265 nm |
|
Alizarin |
430 nm |
|
4-Hydroxybenzoic Acid |
278 nm |
|
Salicylic acid |
278 nm |
Q5. A chromatogram of the sample should be shown
R5. The chromatogram was given in manuscript.
Fig. 1. The chromatogram for phenolic acids in CT by the LC-MS/MS method.
Q6. The antioxidant activity of the sample, as measured by the applied methods, is negligible. This must be clear in all parts of the manuscript
R6. This part was revised.
Q7. In general, the paper does not contain a discussion (comparison with other works) related with the same kind of samples/plant or the same botanical species.
R7. Necessary additions were added to the manuscript in line with the referee's recommendation and highlighted in yellow.

Reviewer 2 Report
In introduction, mention the total average production of maize grain and ranking in production to highlight the importance of crop as well as by-product.
Authors can use pictographic view to explain physiology of corn and indicate which part is called as corn tassel. Also mention how much portion of corn constitutes a tassel in percentage or by weight.
Authors already explained in introduction that presence of various bioactive compounds in corn tassel is reported by various scientists, have they also quantified these compounds? If yes, what is need of taking new research for this?
Knowledge gap and need of research is not critically justified. Authors should elaborate this part.
Line 72: Write it as “R2”. 2 should be at superscript.
Line 78: “higher than 0.99”, write exact value.
It is mentioned in the Statistical Analysis that “Differences between the data were given statistically significant at a p-value ≤ 0.05.” However, present research is only about quantification of bioactive compounds and there is no use of different treatments. Therefore it is not clear which data was evaluated using ANOVA. Also it is nowhere explained in results and discussion.
Author Response
Manuscript number: plants-1799125
MS Type: Full Paper
Title: Phenolic Profile Antioxidant, Anticholinergic, and Antibacterial Properties of Corn Tassel
Correspondence Authors: Prof. Dr. Jameel M. Al-Khayr and Assoc. Prof. Arzu Kavaz Yüksel
Response to Reviewer Comments
First, I want to thank you for your informative and necessary comments on my manuscript. I have tried to do necessary modifications in light of your recommendations accordingly and detailed corrections are listed below point by point.
All manuscript was revised based on the editor and reviewer’s comments. The corrections were done in the text according to the reviewer’s suggestion. The changes made during the manuscript revision have been highlighted with a yellow background for the REVIEWER1 comment; while REVIEWER 2's comments were highlighted with a green background and general changes were highlighted in gray.
We appreciate your time and recommendations to improve this paper.
Thank you for your interest.
Reviewer #2:
Q1. In the introduction, mention the total average production of maize grain and ranking in production to highlight the importance of the crop as well as its by-product.
R1. Corn (Zea mays L.), is one of the most produced cereals in the world. The various uses of corn as food, fodder, animal feed, and lately biofuel have further made it a more favorable and valuable crop. Worldwide, on 197.20 million hectares of land, 1148.48 million metric tonnes of maize were harvested, on average [1].
Corn kernels are important material of the corn plant and are used as a raw material in starch and flour making, production of animal feed, and industrial products such as bioethanol, enzymes, vitamins, organic acids, etc. Other important waste products derived from the corn plant are corn tassel, kernel, and corn cobs which are good sources of bioactive compounds flavonoids, phenolic compounds, anthocyanins, proteins, vitamins, minerals, carotenoids, and volatile compounds [2–5].
The wanted revision was made in the introduction about the total average production of maize grain and ranking in production to highlight the importance of the crop as well as its by-product.
Q2. Authors can use pictographic view to explain the physiology of corn and indicate which part is called as corn tassel. Also mention how many portions of corn constitutes a tassel in percentage or by weight.
R2. Corn tassel (CT) takes place at the top of the corn plant as a male floral organ and produces pollens being carried by the wind to pollinate female flowers named silk. Tassel may produce an average of 25 million pollen grains, on average [6]. Pollens have a high potential for health and are a good source of nutrients and antioxidants, protein, oils, and sugar [7].
An average of 25-30 g of the fresh tassel is obtained from a corn plant and its weight after drying is 8-9 g at most. The weight of tassels in corn depended on the structure of tassels in different corn varieties, especially the number of tassel branches, the length of anthers and tassels, and other factors [5].
The requested information has been added to the introduction.
Q3. Authors already explained in the introduction that the presence of various bioactive compounds in corn tassel is reported by various scientists, have they also quantified these compounds? If yes, what is needed of taking new research for this?
R3. In previous studies, the phenolic acid content of natural-origin CT has not been studied. On the other hand, previous studies mostly focused on the antibacterial effect of corn silk extracts, while antibacterial activity against the CT extracts was not evaluated [29,30]. Also, there is no study about the determination of some bioactive compound profiles and antibacterial properties of CT grown in the Trabzon region. This study aims to show phenolic compound composition and some other important properties of CT grown in Trabzon, Turkey. Also, CT’s antioxidant, anticholinergic, and antibacterial properties were determined using different types of methods. In addition, this study aims to bring together some beneficial information about this study. The antioxidants and phytochemicals that are found in CT were used as functional food supplements, antibacterial products, natural pharmaceuticals, food additives, and also cosmetic products.
Q4. Knowledge gap and need of research is not critically justified. Authors should elaborate this part.
R4. The requested changes were made in the manuscript.
Q5. Line 72: Write it as “R2”. 2 should be a superscript.
R5. The Wanted revision was made and highlighted in green.
Q6. Line 78: “higher than 0.99”, write exact value.
R6. The wanted revision was made” The correlation coefficients ( R2 ) of the standards were found around 0.994-1.011.”
Q7. It is mentioned in the Statistical Analysis that “Differences between the data were given statistically significant at a p-value ≤ 0.05.” However, present research is only about quantification of bioactive compounds and there is no use of different treatments. Therefore it is not clear which data was evaluated using ANOVA. Also it is nowhere explained in results and discussion.
R7. Statistical Analysis
The analysis results were evaluated with GraphPad Prism version 8 (GraphPad Software, La Jolla, California, USA). The results were shown as 95% confidence intervals (mean±standard deviation).
The wanted revisions were made to the manuscript.

Round 2
Reviewer 1 Report
THe authirs have significantly improved the manuscript. Still Fig 1 is confusing.... please improve it
As Figure 1 represents the LC-MS chromatogram of CT sample, it should only be 1 line in the graphic (there are several lines in distinct colours).
Author Response
Manuscript number: plants-1799125
MS Type: Full Paper
Title: Phenolic Profile Antioxidant, Anticholinergic, and Antibacterial Properties of Corn Tassel
Correspondence Authors: Prof. Dr. Jameel M. Al-Khayri and Assoc. Prof. Arzu Kavaz Yüksel
Response to Reviewer Comments
First, I want to thank you for your informative and necessary comments on my manuscript. I have tried to do necessary modifications in light of your recommendations accordingly and detailed corrections are listed below point by point.
All manuscript was revised based on the editor and reviewer’s comments. The corrections were done in the text according to the reviewer’s suggestion. The changes made during the manuscript revision have been highlighted with a yellow background for the REVIEWER1 comment and general changes were highlighted in gray.
We appreciate your time and recommendations to improve this paper.
Thank you for your interest.
Reviewer #1:
Q1. THe authors have significantly improved the manuscript. Still, Fig 1 is confusing.... please improve it
R1. Figure 1. was revised.
Figure 1. The chromatogram of the phenolic acids in CT extract obtained by the LC-MS/MS method

Reviewer 2 Report
Thanks for revising the article.
Author Response
Manuscript number: plants-1799125
MS Type: Full Paper
Title: Phenolic Profile Antioxidant, Anticholinergic, and Antibacterial Properties of Corn Tassel
Correspondence Authors: Prof. Dr. Jameel M. Al-Khayri and Assoc. Prof. Arzu Kavaz Yüksel
Response to Reviewer Comments
First, I want to thank you for your informative and necessary comments on my manuscript. I have tried to do necessary modifications in light of your recommendations accordingly and detailed corrections are listed below point by point.
All manuscript was revised based on the editor and reviewer’s comments. The corrections were done in the text according to the reviewer’s suggestion. The changes made during the manuscript revision have been highlighted with a yellow background for the REVIEWER1 comment and general changes were highlighted in gray.
We appreciate your time and recommendations to improve this paper.
Thank you for your interest.
